# Asymmetric distribution of color-opponent response types across mouse visual cortex supports superior color vision in the sky

Katrin Franke[1,2,3,4*], Chenchen Cai[5,6], Kayla Ponder[4], Jiakun Fu[4], Sacha Sokoloski[5,7], Philipp Berens[5,7], Andreas Savas Tolias[1,2,3,4,8]

[1]Department of Ophthalmology, Byers Eye Institute, Stanford University School of Medicine, Stanford, United States; [2]Stanford Bio-X, Stanford University, Stanford, United States; [3]Wu Tsai Neurosciences Institute, Stanford University, Stanford, United States; [4]Department of Neuroscience & Center for Neuroscience and Artificial Intelligence, Baylor College of Medicine, Houston, United States; [5]Institute for Ophthalmic Research, University of Tübingen, Tübingen, Germany; [6]Graduate Training Center of Neuroscience, International Max Planck Research School, University of Tübingen, Tübingen, Germany; [7]Hertie Institute for AI in Brain Health, University of Tübingen, Tübingen, Germany; [8]Department of Electrical Engineering, Stanford University, Stanford, United States

**\*For correspondence:**
kafranke@stanford.edu

**Competing interest:** The authors declare that no competing interests exist.

**Abstract** Color is an important visual feature that informs behavior, and the retinal basis for color vision has been studied across various vertebrate species. While many studies have investigated how color information is processed in visual brain areas of primate species, we have limited understanding of how it is organized beyond the retina in other species, including most dichromatic mammals. In this study, we systematically characterized how color is represented in the primary visual cortex (V1) of mice. Using large-scale neuronal recordings and a luminance and color noise stimulus, we found that more than a third of neurons in mouse V1 are color-opponent in their receptive field center, while the receptive field surround predominantly captures luminance contrast. Furthermore, we found that color-opponency is especially pronounced in posterior V1 that encodes the sky, matching the statistics of natural scenes experienced by mice. Using unsupervised clustering, we demonstrate that the asymmetry in color representations across cortex can be explained by an uneven distribution of green-On/UV-Off color-opponent response types that are represented in the upper visual field. Finally, a simple model with natural scene-inspired parametric stimuli shows that green-On/UV-Off color-opponent response types may enhance the detection of 'predatory'-like dark UV-objects in noisy daylight scenes. The results from this study highlight the relevance of color processing in the mouse visual system and contribute to our understanding of how color information is organized in the visual hierarchy across species.

## eLife assessment

Franke et al. explore and characterize color response properties of neurons in mouse primary visual cortex (V1), revealing specific color opponent encoding strategies across the visual field. The paper provides evidence for the existence of color opponency in a subset of neurons within V1 and shows that these color opponent neurons are more numerous in the upper visual field. Support for the

main conclusions is **convincing** and the dataset that forms the basis of the paper is impressive. The paper will make an **important** contribution to understanding how color is coded in mouse V1.

## Introduction

Color is an important property of the visual world informing behavior. The retinal basis for color vision has been studied in many vertebrate species, including zebrafish, mice, and primates (reviewed in *Baden and Osorio, 2019*): Signals from different photoreceptor types which are sensitive to different wavelengths are compared by retinal circuits, thereby creating color-opponent cell types. In primate species, it is well studied how color-opponent signals from the retina are processed in downstream brain areas (*Livingstone and Hubel, 1984*; *Wiesel and Hubel, 1966*; *Gegenfurtner et al., 1996*; *Tanigawa et al., 2010*; *Chatterjee and Callaway, 2003*). In most other species, however, we know relatively little about how color information is processed beyond the retina. Thus, our understanding of color processing along the visual hierarchy across species remains limited, highlighting the need for further research to uncover general rules governing this fundamental aspect of vision.

Here, we systematically studied how color is represented in the primary visual cortex (V1) of mice. Like most mammals, mice are dichromatic and have two cone photoreceptor types, expressing ultra-violet (UV)- and green-sensitive S- and M-opsin (*Szél et al., 1992*), respectively. In addition, they have one type of rod photoreceptor which is green-sensitive. Importantly, UV- and green-sensitive cone photoreceptors predominantly sample the upper and lower visual field, respectively, through an uneven opsin distribution across the retina (*Szél et al., 1992*; *Baden et al., 2013*). Behavioral studies have demonstrated that mice can discriminate different colors (*Jacobs et al., 2004*), at least in the upper visual field (*Denman et al., 2018*). However, a thorough understanding of the neuronal correlates underlying this behavior is still missing.

At the level of the mouse retina, a large body of literature has identified mechanisms underlying color-opponent responses, including cone-type selective (*Stabio et al., 2018*; *Nadal-Nicolás et al., 2020*; *Haverkamp et al., 2005*) or cone-type unselective wiring (*Chang et al., 2013*) and rod-cone opponency (*Joesch and Meister, 2016*; *Szatko et al., 2020*; *Khani and Gollisch, 2021*). The latter is widespread across many neuron types located in the ventral retina sampling the sky, where rod and cone photoreceptors exhibit the strongest difference in spectral sensitivity, and requires the integration across center and surround components of receptive fields (RFs). In visual areas downstream to the retina, the frequency of color-opponency has remained controversial. Some studies have reported very low numbers of color-opponent neurons in mouse dLGN (*Denman et al., 2017*) and V1 (*Tan et al., 2015*), while two more recent studies identified pronounced cone- and rod-cone-dependent color-opponency (*Mouland et al., 2021*; *Rhim and Nauhaus, 2023*).

In this study, we systematically characterized color and luminance center-surround RF properties of mouse V1 neurons across different light levels using large-scale neuronal recordings and a luminance and color noise stimulus. This revealed that more than a third of neurons in mouse V1 are highly sensitive to color features of the visual input in their RF center, while the RF surround predominantly captures luminance contrast. Color-opponency in the RF center was strongest for photopic light levels largely activating cone photoreceptors and greatly decreased for mesopic light levels, suggesting that the observed color-opponency in V1 is at least partially mediated by the comparison of cone photoreceptor signals. We further showed that color-opponency is especially pronounced in posterior V1 which encodes the sky, in line with previous work in the retina (*Szatko et al., 2020*), and matching the statistics of mouse natural scenes (*Qiu et al., 2021*; *Abballe and Asari, 2022*). Using unsupervised clustering we demonstrated that the asymmetry in color representations across cortex can be explained by an uneven distribution of green-On/UV-Off color-opponent response types that almost exclusively represented the upper visual field. Finally, by implementing a simple model with natural scene inspired parametric stimuli, we showed that green-On/UV-Off color-opponent response types may enhance the detection of 'predatory'-like dark UV-objects in noisy daylight scenes.

The results of our study support the hypothesis that neurons in the visual cortex asymmetrically represent information across the visual field, facilitating specific visual tasks such as the robust detection of aerial predators in noisy natural scenes.

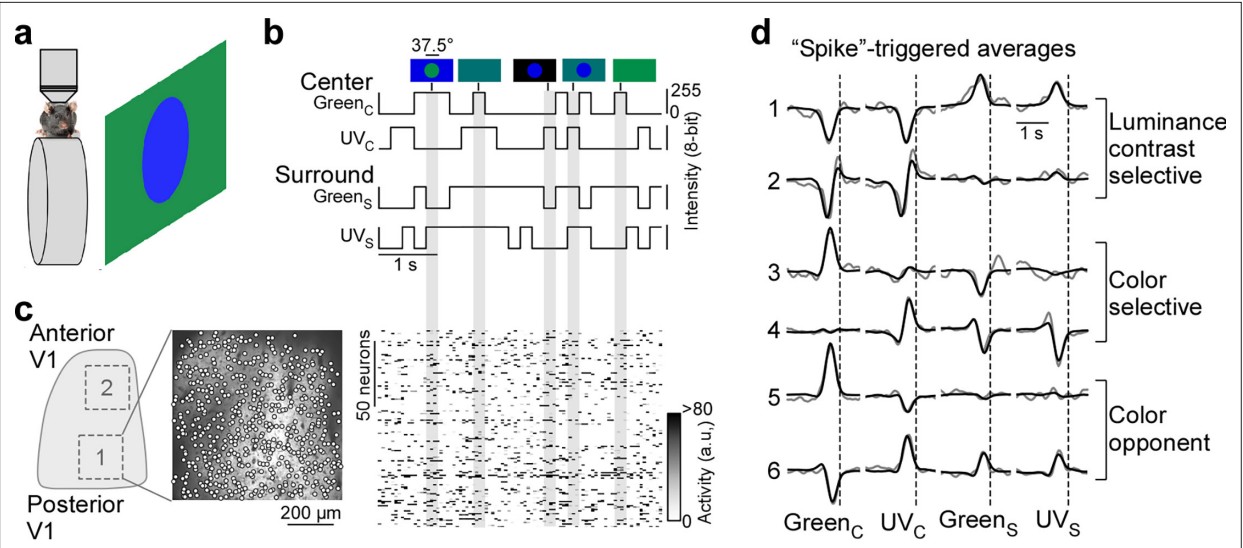

**Figure 1.** Color noise stimulus identifies center-surround receptive field properties of mouse primary visual cortex (V1) neurons. (**a**) Schematic illustrating experimental setup: Awake, head-fixed mice on a treadmill were presented with a center-surround color noise stimulus while recording the population calcium activity in L2/3 neurons of V1 using two-photon imaging. Stimuli were back-projected on a Teflon screen by a DLP-based projector equipped with a ultraviolet (UV) (390 nm) and green (460 nm) LED, allowing to differentially activate mouse cone photoreceptors. (**b**) Schematic drawing illustrating stimulus paradigm: UV and green center spot (UVC /GreenC) and surround annulus (UVS /Green$_S$) flickered independently at 5 Hz according to binary random sequences. Top images depict example stimulus frames. See also *Figure 1—figure supplement 1*. (**c**) Left side shows a schematic of V1 with a posterior and anterior recording field, and the recorded neurons of the posterior field overlaid on top of the mean projection of the recording. Right side shows the activity of *n*=150 neurons of this recording in response to the stimulus sequence shown in (**b**). (**d**) Event-triggered averages (ETAs) of six example neurons, shown for the four stimulus conditions. Gray: Original ETA. Black: Reconstruction using principal component analysis (PCA). See also *Figure 1—figure supplement 2*. Cells are grouped based on their ETA properties and include luminance-sensitive, color selective, and color-opponent neurons. Black dotted lines indicate time of response.

The online version of this article includes the following figure supplement(s) for figure 1:

**Figure supplement 1.** Verification of stimulus paradigm.

**Figure supplement 2.** Event-triggered averages (ETAs) and quality control.

**Figure supplement 3.** Reconstruction of event-triggered averages (ETAs) using sparse principal component analysis (PCA).

## Results

### Characterizing color and luminance center-surround RFs of mouse V1 neurons

To study the neuronal representation of color in mouse V1, we characterized center (i.e. classical) and surround (i.e. extra-classical) RFs of excitatory V1 neurons in awake, head-fixed mice in response to a luminance and color noise stimulus (*Figure 1a*). The noise stimulus consisted of a center spot (37.5 degrees visual angle in diameter) and a surround annulus (approx. 120×90 degrees visual angle without the center spot) that simultaneously flickered in UV and green based on 5 Hz binary random sequences (*Figure 1b*), thereby capturing chromatic, temporal, as well as one spatial dimension of the neurons' RFs. Neuronal responses to such relatively simple, parametric stimuli are easy to interpret and allow to systematically quantify chromatic RF properties of visual neurons in the mouse (*Szatko et al., 2020*) and zebrafish retina (*Zimmermann et al., 2018*), as well as in primate V1 (*Chatterjee and Callaway, 2003*). We presented visual stimuli to awake, head-fixed mice positioned on a treadmill while at the same time recording the population calcium activity within L2/3 of V1 using two-photon imaging (700×700 µm² recordings at 15 Hz). Visual stimulation was performed in the photopic light regime that predominantly activates cone photoreceptors. We back-projected visual stimuli on a Teflon screen using a custom projector with UV and green LEDs that allow differential activation of mouse cone photoreceptors (*Franke et al., 2019*; *Franke et al., 2022*). Functional recordings were obtained from posterior and anterior V1 (*Figure 1c*), encoding the upper and lower visual field (*Schuett et al., 2002*), respectively.

To record from many V1 neurons simultaneously, we used a center stimulus size of 37.5 degrees visual angle in diameter, which is slightly larger than the center RFs we estimated for single V1 neurons (26.2±4.6 degrees visual angle in diameter) using a sparse noise paradigm (*Jones and Palmer, 1987*). The disadvantage of this approach is that the stimulus is only roughly centered on the neurons' center RFs. To reduce the impact of potential stimulus misalignment on our results, we used the following steps and controls. First, for each recording, we positioned the monitor such that the population RF across all neurons, estimated using a short sparse noise stimulus, lies within the center of the stimulus field of view. Second, we confirmed that this procedure results in good stimulus alignment at the level of individual neurons by using a longer sparse noise stimulus for a subset of experiments (*Figure 1—figure supplement 1a, b*). Specifically, we found that for the majority of tested neurons (83%), more than two-thirds of their center RF overlapped with the center spot of the color noise stimulus (*Figure 1—figure supplement 1b*). Finally, we excluded neurons from analysis, which did not show significant center responses (*n*=1937 neurons excluded from *n*=5248; *Figure 1—figure supplement 2d, e*), which may be caused by misalignment of the stimulus. Together, this suggests that the center spot and the surround annulus of the noise stimulus predominantly drive center (i.e. classical RF) and surround (i.e. extra-classical RF), respectively, of the recorded V1 neurons.

For analysis of the neurons' stimulus preference, we first deconvolved the calcium traces (*Figure 1—figure supplement 2a*) to account for the slow kinetics of the calcium sensor (e.g. *Pachitariu et al., 2018*) and then used the deconvolved noise responses of each neuron to estimate an 'event-triggered average' (ETAs) for the four stimulus conditions - center (C) and surround (S) for both UV and green (Green$_C$, UV$_C$, Green$_S$, UV$_C$). Specifically, deconvolved neuronal responses were reverse-correlated with the stimulus trace and the raw ETAs were then transformed into a lower dimensional representation using principal component analysis (PCA; *Figure 1d* and *Figure 1—figure supplement 3*). In other words, the ETA was obtained by summing the stimulus sequences that elicit an event (i.e. response), weighted by the amplitude of the response. Consequently, the absolute amplitude of the ETA correlated with the calcium amplitude and the ETA amplitudes of different stimulus conditions were comparable. Note that the deconvolution of raw calcium responses changes the kinetics of the ETAs, but not the neurons' stimulus selectivity (*Figure 1—figure supplement 2b, c*).

Using this approach, we obtained ETAs of *n*=3331 excitatory V1 neurons (*n*=6 recording fields, *n*=3 mice) with diverse center-surround stimulus preferences (*Figure 1d*, *Figure 1—figure supplement 1c*). This included neurons sensitive to luminance contrast that did not discriminate between stimulus color (cells 1 and 2 in *Figure 1d*) and color selective cells only responding to one color of the stimulus (cells 3 and 4). In addition, some neurons exhibited color-opponency in the center (cells 5 and 6) or surround, meaning that a neuron prefers a stimulus of opposite polarity in the UV and green channel (e.g. UV-On and green-Off). To validate our experimental approach, we confirmed that the noise stimulus recovers well-described RF properties of mouse V1 neurons. First, the majority of neurons showed negatively correlated center and surround ETAs for both the UV and green channels (*Figure 1—figure supplement 1d*), meaning that a neuron preferring an increment of light in the center ('On') favors a light decrement in the surround ('Off') and vice versa. This finding is consistent with On-Off center-surround antagonism of neurons in early visual areas, and has been described in both the mouse retina (e.g. *Franke et al., 2017*) and mouse thalamus (e.g. *Grubb and Thompson, 2003*). Second, neurons recorded in posterior and anterior V1 preferred UV and green stimuli (*Figure 1—figure supplement 1e, f*), respectively, in line with the distribution of cone opsins across the retina (*Szél et al., 1992*; *Baden et al., 2013*) and previous cortical work (*Rhim et al., 2017*; *Franke et al., 2022*; *Aihara et al., 2017*). This asymmetry in color preference was less pronounced for the surround (*Figure 1—figure supplement 1e, f*), as has been reported for retinal neurons in mice (*Szatko et al., 2020*). Taken together, these results show that our experimental paradigm using a parametric luminance and color noise stimulus accurately captures known center-surround RF properties of cortical neurons in mice.

## Color contrast is represented by the RF center in a large number of mouse V1 neurons

To systematically study how color is represented by the population of mouse V1 neurons, we mapped each cell's center and surround ETA into a two-dimensional space depicting neuronal sensitivity for luminance and color contrast (*Figure 2a*). For each neuron, we extracted ETA peak amplitudes relative

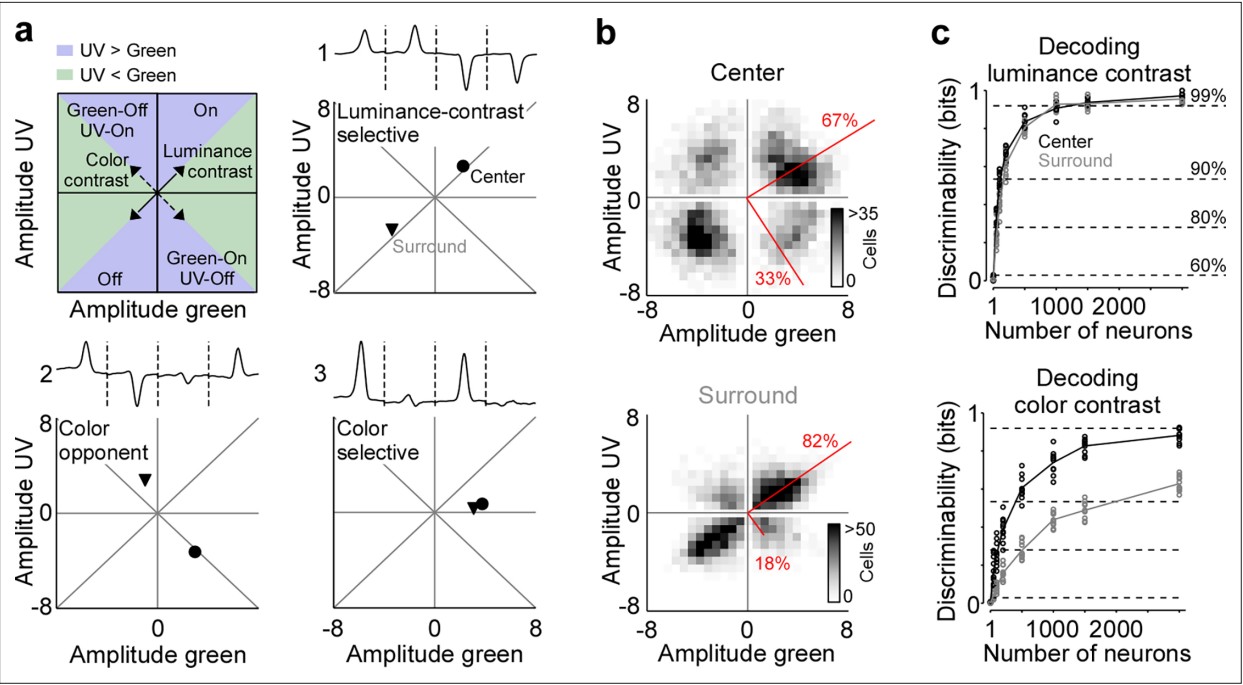

**Figure 2.** Strong neuronal representation of color in mouse primary visual cortex. (**a**) Top left panel shows schematic drawing illustrating the green and ultraviolet (UV) contrast space used in the other panels and *Figures 3–5*. Amplitudes below and above zero indicates an Off and On cell, respectively. Achromatic On and Off cells will scatter in the lower left and upper right quadrants along the diagonal ('luminance contrast'), while color-opponent cells will fall within the upper left and lower right quadrants along the off-diagonal ('color contrast'). Blue and green shading indicates stronger responses to UV and green stimuli, respectively. The three other panels show event-triggered averages (ETAs) of three example neurons (top) with the peak amplitudes ('contrast') of their center (dot) and surround responses (triangle) indicated in the bottom. (**b**) Density plot of peak amplitudes of center (top) and surround (bottom) ETAs across all neurons (*n*=3331 cells, *n*=6 recording fields, *n*=3 mice). Red lines correspond to axes of principal components (PCs) obtained from a principal component analysis (PCA) on the center or surround data, with percentage of variance explained along the polarity and color axis indicated. For reproducibility across animals, see *Figure 2—figure supplement 1*. The percentages of variance explained by color (off-diagonal) and luminance axis (diagonal) correlate with the number of neurons located in the color (top left and bottom right) and luminance contrast quadrants (top right and bottom left), respectively. Scale bars indicate the number of neurons in the 2D histogram. (**c**) Decoding discriminability of stimulus luminance (top) and stimulus color (bottom) based on center (black) and surround (gray) responses of different numbers of neurons. Decoding was performed using a support vector machine (SVM). Lines indicate the mean of 10-fold cross-validation (shown as dots). For luminance contrast, decoding discriminability was significantly different between center and surround for *n*=50 and *n*=100 neurons (t-test for unpaired data, p-value was adjusted for multiple comparisons using Bonferroni correction). For color contrast, decoding discriminability was significantly different between center and surround for all numbers of neurons tested, except *n*=1 neuron. Dotted horizontal lines indicate decoding accuracy in % for 60%, 80%, 90%, and 99%, with a change level of 50% corresponding to 0 bits.

The online version of this article includes the following figure supplement(s) for figure 2:

**Figure supplement 1.** Consistency across mice.

**Figure supplement 2.** Neuronal representation of color in the mouse retina.

to baseline for all four stimulus conditions, with positive and negative peak amplitudes for On and Off stimulus preference, respectively. In this space, neurons sensitive to luminance contrast responding with the same polarity (i.e. On versus Off) to either color of the stimulus fall along the diagonal (cell 1 in *Figure 2a*) and color-opponent neurons scatter along the off-diagonal (cell 2). In addition, the neuronal selectivity for UV and green stimuli is indicated by the relative distance to the *x*- and *y*-axis (cell 3), respectively. We found that most V1 neurons were sensitive to luminance contrast and fell in the upper right or lower left quadrant along the diagonal, both for the center and surround component of V1 RFs (*Figure 2b*). Nevertheless, a substantial fraction of neurons (33.1%) preferred color-opponent stimuli and scattered along the off-diagonal in the upper left and lower right quadrants, especially for the RF center. We quantified the fraction of variance explained by the luminance versus the color axis across the neuronal population by performing PCA on the center and surround contrast space, respectively (*Qiu et al., 2021*). The luminance axis captured the major part of the variance of stimulus sensitivity for the RF surround (82%), while it explained less of the variance for the RF center

(67%). As a result, one-third (33%) of the variance within the tested stimulus sensitivity space of the RF center was explained by the color axis. Please note that the percentages of variance explained by color and luminance axis correlate with the number of neurons located in the color (top left and bottom right) and luminance contrast quadrants (top right and bottom left), respectively. Our results were consistent for a more conservative quality threshold, which only considered the best 25% of neurons (*Figure 1—figure supplement 2e, f*). In addition, the above results obtained by pooling data across animals were consistent within all three mice tested (*Figure 2—figure supplement 1a*). Therefore, all the following analyses were based on data pooled across animals.

We confirmed and quantified the pronounced color-opponency in mouse V1 we observed based on the neurons' preferred stimuli (i.e. ETAs) using an independent decoding analysis. For that, we trained a nonlinear support vector machine (SVM) to decode stimulus luminance (On versus Off) or color (UV versus green) based on the recorded and deconvolved neuronal activity. Decoding was performed using 10-fold cross-validation and decoding accuracy (in %) was transformed into mutual information (in bits). The discriminability of luminance contrast rapidly increased with the number of neurons used by the SVM decoder, and saturated close to perfect discriminability (>0.92 bits, i.e. >99% accuracy) when including more than 1000 neurons (*Figure 2c*). Interestingly, decoding performance was similar for center and surround stimuli, indicating that V1 responses are as informative about luminance contrast in the center as in the surround. We next trained a decoder to discriminate stimulus color. The decoding performance was lower for stimulus color compared to stimulus luminance (*Figure 2c*), consistent with the finding described above that V1 neurons are more sensitive to luminance than color contrast. In addition, discriminability of stimulus color was significantly better for stimuli presented in the RF center compared to stimuli shown in the RF surround, thereby verifying our ETA results. Together, our results demonstrate that for photopic light levels, neurons in mouse visual cortex strongly encode color features of the visual input in their RF center, while the RF surround predominantly captures luminance contrast.

Next, we tested to what extent the strong representation of color by the center component of V1 RFs was inherited by color-opponency present in the center RF of retinal output neurons (*Szatko et al., 2020*; *Khani and Gollisch, 2021*). We tested this by using a publicly available dataset of retinal ganglion cell responses to a center and surround color flicker stimulus (*Szatko et al., 2020*), similar to the one used here, but with center and surround stimulus sizes adjusted to the smaller RF sizes of retinal neurons. We embedded each cell's center and surround ETA into the luminance and color contrast space described above (*Figure 2—figure supplement 2*). We found that at the level of the retinal output, the color axis explained only 12% of the variance in the tested sensitivity space for the RF center (10 degrees visual angle), which is much lower than what we observed for V1 center RFs (37.5 degrees visual angle). The low fraction of center color-opponent retinal ganglion cells is in line with a recent study that characterized RF properties of these neurons using natural movies recorded in the mouse's natural environment (*Hoefling et al., 2022*). Collectively, these findings suggest that the pronounced center color-opponency in V1 neurons cannot be solely attributed to the color-opponency present in the RF center of retinal ganglion cells. It likely depends on the activation of both the center and surround of retinal neurons, as well as a potential remapping of retinal center and surround RFs in downstream processing stages.

## The neuronal representation of color in mouse V1 decreases with lower ambient light levels

Previous studies have reported varying numbers of color-opponent neurons in mouse visual areas, ranging from very few in mouse dLGN (*Denman et al., 2017*) and V1 (*Tan et al., 2015*) to a large number in the thalamus (*Mouland et al., 2021*), visual cortex (*Rhim and Nauhaus, 2023*), and the retina (*Szatko et al., 2020*). In part, this discrepancy regarding the role of color for visual processing in mice is likely due to the fact that different studies have used different light levels, resulting in varying activations of rod and cone photoreceptors that are both involved in chromatic processing. To systematically study how ambient light levels affect the neuronal representation of color in mouse visual cortex, we repeated our experiments with the noise stimulus performed in photopic conditions (approx. 15,000 photoisomerizations (P*) per cone and second) in high (approx. 400 P* per cone and second) and low mesopic light conditions (approx. 50 P* per cone and second). The high mesopic light condition is expected to equally activate rod and cone photoreceptors, while the low mesopic

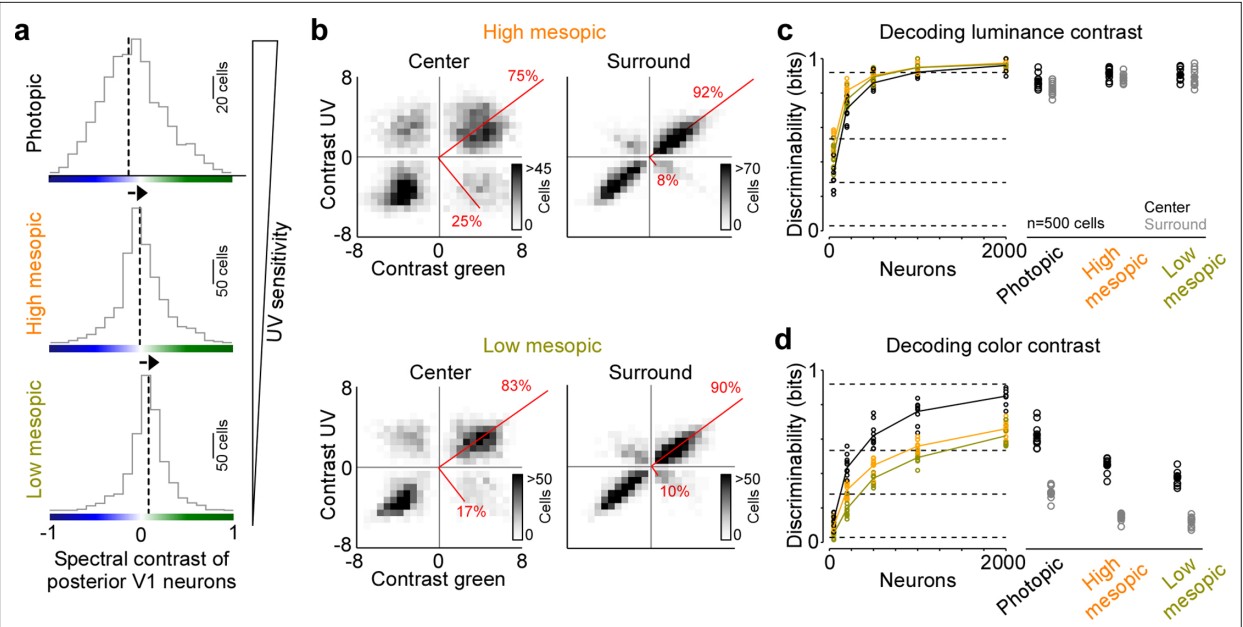

**Figure 3.** Reduced representation of color contrast in mouse primary visual cortex (V1) for lower ambient light levels. (**a**) Distribution of spectral contrast of center event-triggered averages (ETAs) of all neurons recorded in posterior V1, for photopic (top, n=1616 cells, n=3 recording fields, n=3 mice), high mesopic (middle, n=1485 cells, n=3 recording fields, n=3 mice), and low mesopic (bottom, n=1295 cells, n=3 recording fields, n=3 mice) ambient light levels. Black dotted lines indicate mean of distribution. Spectral contrast significantly differed across all combinations of light levels (t-test for unpaired data, p-value was adjusted for multiple comparisons using Bonferroni correction). The triangle on the right indicates ultraviolet (UV) sensitivity of the neurons, which is decreasing with lower ambient light levels. (**b**) Density plot of peak amplitudes of center (left) and surround (right) ETAs. Red lines correspond to axes of principal components (PCs) obtained from a principal component analysis (PCA) on the center or surround data, with percentage of variance explained along the polarity and color axis indicated. Top row shows high mesopic (n=3522 cells, n=6 recording fields, n=3 mice) and bottom row low mesopic (n=2705 cells, n=6 recording fields, n=3 mice) light levels. The percentages of variance explained by color (off-diagonal) and luminance axis (diagonal) correlate with the number of neurons located in the color (top left and bottom right) and luminance contrast quadrants (top right and bottom left), respectively. Scale bars indicate the number of neurons in the 2D histogram. (**c**) Discriminability (in bits) of luminance contrast (On versus Off) for the center across the three light levels tested, obtained from training support vector machine (SVM) decoders based on recorded noise responses of V1 neurons. Right plot shows the discriminability of luminance contrast for n=500 neurons for center and surround. Dots show decoding performance of 10 train/test trial splits. For n=500 neurons, decoding discriminability of the center was not significantly different across light levels (t-test for unpaired data, p-value was adjusted for multiple comparisons using Bonferroni correction). The surround discriminability was significantly lower than the center for the photopic condition. Dotted horizontal lines indicate decoding accuracy in % for 60%, 80%, 90%, and 99%, with a change level of 50% corresponding to 0 bits. (**d**) Like (**c**), but showing discriminability of color contrast (green versus UV). Decoding discriminability was significantly different between center and surround for all three light levels. In addition, discriminability for the center was significantly different between photopic and mesopic conditions, but not between the two mesopic conditions (t-test for unpaired data, p-value was adjusted for multiple comparisons using Bonferroni correction). Dotted horizontal lines indicate decoding accuracy in % for 60%, 80%, 90%, and 99%, with a change level of 50% corresponding to 0 bits.

condition largely drives rod photoreceptors, with only a small cone contribution. Indeed, decreasing ambient light levels resulted in a reduction of UV sensitivity in posterior V1 neurons (*Figure 3a*), indicative for a gradual activation shift from UV-sensitive cones to rods, which are green sensitive. The neuronal representation of color greatly decreased when reducing ambient light levels: Both the fraction of ETA variance explained by the color axis (*Figure 3b*) and the decoding discriminability of stimulus color dropped significantly (*Figure 3d*). The drop in decoding discriminability was not due to lower signal-to-noise levels for the mesopic light levels, as the decoding of stimulus luminance was not significantly higher for the photopic condition compared to the mesopic conditions (*Figure 3c*). Across all light levels tested, the RF surround of V1 neurons was less informative about stimulus color, resulting in 40–60% lower discriminability in the surround compared to the center (*Figure 3d*). Interestingly, even for the lowest light level, where only a small fraction of ETA variance was explained by stimulus color (*Figure 3b*), V1 neurons reliably encoded color information (*Figure 3d*). This suggests that weak cone activation in low mesopic light levels is sufficient to extract color features from the visual input. In summary, our results demonstrate that the neuronal representation of color in mouse

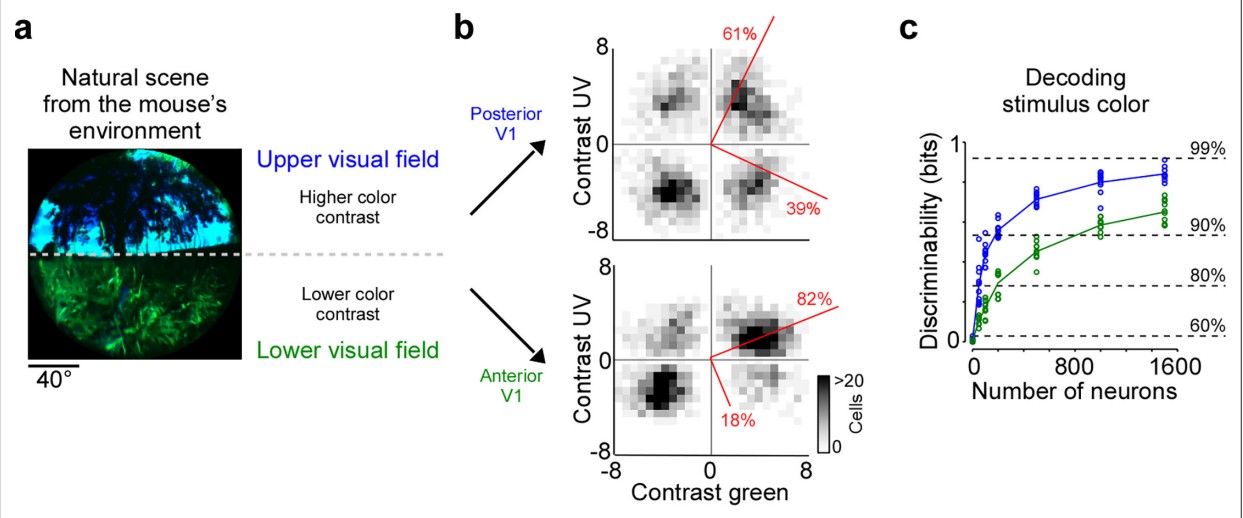

**Figure 4.** Cortical representation of color changes across visual space. (**a**) Natural scene captured in the natural environment of mice using a custom-built camera adjusted to the mouse's spectral sensitivity (*Qiu et al., 2021*). Dashed line indicates the horizon and separates the scene into lower and upper visual field. Previous studies *Qiu et al., 2021*; *Abballe and Asari, 2022* have reported higher color contrast in the upper compared to the lower visual field. (**b**) Density plot of peak amplitudes of center event-triggered averages (ETAs) across neurons recorded in posterior (top) and anterior primary visual cortex (V1) (bottom). Red lines correspond to axes of principal components (PCs) obtained from a principal component analysis (PCA), with percentage of variance explained along the polarity and color axis indicated. For reproducibility across animals, see *Figure 2—figure supplement 1*. The percentages of variance explained by color (off-diagonal) and luminance axis (diagonal) correlate with the number of neurons located in the color (top left and bottom right) and luminance contrast quadrants (top right and bottom left), respectively. Scale bars indicate the number of neurons in the 2D histogram. (**c**) Discriminability (in bits) of color contrast (ultraviolet [UV] or green) for neurons recorded in posterior (blue) and anterior V1 (green), obtained from training support vector machine (SVM) decoders based on recorded noise responses of V1 neurons. The decoding discriminability was significantly different between anterior and posterior neurons (t-test for unpaired data, p-value was adjusted for multiple comparisons using Bonferroni correction). Dotted horizontal lines indicate decoding accuracy in % for 60%, 80%, 90%, and 99%, with a change level of 50% corresponding to 0 bits.

visual cortex greatly depends on ambient light levels, and is strongest for photopic light levels that predominantly drive cone photoreceptors.

## Cortical representation of color changes across the visual field

Chromatic and achromatic features present in natural scenes systematically vary across the visual field, with notable differences between regions below and above the horizon (*Nilsson et al., 2022*; *Qiu et al., 2021*). Recently, it has been demonstrated that color contrast in scenes from the mouse's natural environment is enriched in the upper visual field (*Figure 4a*; *Qiu et al., 2021*; *Abballe and Asari, 2022*). To encode the sensory input efficiently, these scene statistics should ideally be reflected in the neuronal representations, as has been observed at the level of the mouse retina (*Szatko et al., 2020*; *Khani and Gollisch, 2021*). To study how the representation of color changes across the visual field in mouse V1, we separately analyzed the neurons recorded in posterior and anterior V1, which encode visual information from the upper and lower visual field, respectively. We focused this analysis on the RF center because V1 surround RFs were on average predominantly explained by luminance contrast (*Figure 2*). We found that the color axis explained twice as much ETA variance in posterior compared to anterior V1 (*Figure 4b*): It captured 39% of the variance in the upper visual field and only 19% of the variance in the lower visual field. This finding was consistent across animals (*Figure 2—figure supplement 1b*). In line with this, the discriminability of stimulus color was significantly higher when using the responses of posterior V1 neurons for decoding (*Figure 4c*). Together, this revealed a stronger cortical representation of color in posterior than anterior mouse visual cortex, which might be an adaptation to efficiently encode the enriched color contrast in the upper visual field of mouse natural scenes (*Qiu et al., 2021*).

## Asymmetric distribution of color response types explains higher color sensitivity in posterior V1

Next, we investigated the mechanism underlying this asymmetry in color encoding across mouse visual cortex. In the mouse retina, different retinal ganglion cell types are differentially distributed across the retina and, therefore, asymmetrically sample the visual space (reviewed in *Baden et al., 2020*). For example, W3 cells that have been linked to aerial predator detection exhibit the highest density in the ventral retina looking at the sky (*Zhang et al., 2012*). Similarly, we hypothesized that the difference in decoding performance of stimulus color in posterior and anterior V1 might be due to an asymmetric distribution of functional neuron types sensitive to color versus luminance contrast. To test this, we clustered the ETAs of all neurons into 'functional response types' and quantified the distribution of the identified response types across cortical position. Specifically, we used the features extracted from the ETAs by PCA (*Figure 1—figure supplement 3*) and clustered the feature weights into 17 response types using a Gaussian mixture model (GMM; *Figure 5a* and *Figure 5—figure supplement 1*). We used 17 components for the GMM because this resulted in the best model on held-out test data (*Figure 5—figure supplement 1a*), although the performance was relatively flat for a wide range of components. The mean assignment accuracy of generated ground-truth labels was 89.2% (±6%) and all response types were present in all mice (*Figure 5—figure supplement 1b, c*), indicating that the response types are well separated and robust. The response types greatly differed with respect to functional properties, such as color-opponency, response polarity, and surround antagonism (*Figure 5a*), and, therefore, covered distinct sub-spaces of the color and luminance sensitivity space (*Figure 5—figure supplement 1d*). Approximately half of the response types were sensitive to luminance contrast (types 1–8) and exhibited different response polarities and surround strengths. The other half consisted of types with a strong selectivity for UV or green center stimuli (types 9–13) and color-opponency in the center (types 14–17).

We next investigated the distribution of individual response types across anterior and posterior V1 by computing a cortical distribution index (*Figure 5b*). This index was –1 and 1 if all cells of one response type were located in posterior and anterior V1, respectively, and 0 if the respective response type was evenly distributed across cortex. We found that approximately half of the response types were equally distributed in mouse V1 (distribution index from –0.3 to 0.3), including mostly response types sensitive to luminance contrast. Interestingly, response types with green-Off/UV-On color-opponency were also uniformly spread across the anterior-posterior axis of mouse V1, suggesting that a neuronal substrate supporting color vision exists in both the upper and lower visual field. Response types enriched in anterior V1 (distribution index >0.3) fell along the luminance contrast axis but showed a preference for green center stimuli, consistent with the higher green sensitivity of cone photoreceptors sampling the ground (*Baden et al., 2013*). Similarly, as expected from the high density of UV-sensitive cone photoreceptors in the ventral retina (*Baden et al., 2013*), one response type strongly enriched in the posterior cortex (distribution index <0.3) preferred UV in the RF center. To our surprise, response types with a green-On/UV-Off color-opponency were almost exclusively confined to posterior V1. As a result of this, the color axis explained 73% of ETA variance for the response types enriched in posterior cortex, while it explained only 17% for the anterior-enriched response types. We confirmed the higher sensitivity for color versus luminance contrast of posterior response types by showing that their decoding discriminability of color was significantly better than that for anterior response types (*Figure 5d*). Together, these results demonstrate that the asymmetry in neuronal color tuning across cortical position we report in mice can be explained by an uneven distribution of color-opponent response types.

We next speculated about the computational role of the green-On/UV-Off color-opponent response types largely present in posterior V1. As most predators are expected to approach the mouse from above, color-opponency in the upper visual field could well support threat detection. Especially for visual scenes with inhomogeneous illumination (e.g. in the forest), which result in large intensity fluctuations at the photoreceptor array, color-opponent RF structures may result in a more reliable signal (discussed in *Maximov, 2000*; *Kelber et al., 2003*). To test this prediction, we used parametric stimuli inspired by noisy natural scenes, containing only noise, or a dark ellipse of varying size, angle, and position on top of noise (*Figure 5e*). The dark object had a higher contrast in the UV than green channel, as it has been shown that objects in the sky (*Qiu et al., 2021*), underwater (*Cronin and Bok, 2016*; *Losey et al., 1999*), or in the snow (*Tyler et al., 2014*) are often more visible in the UV than

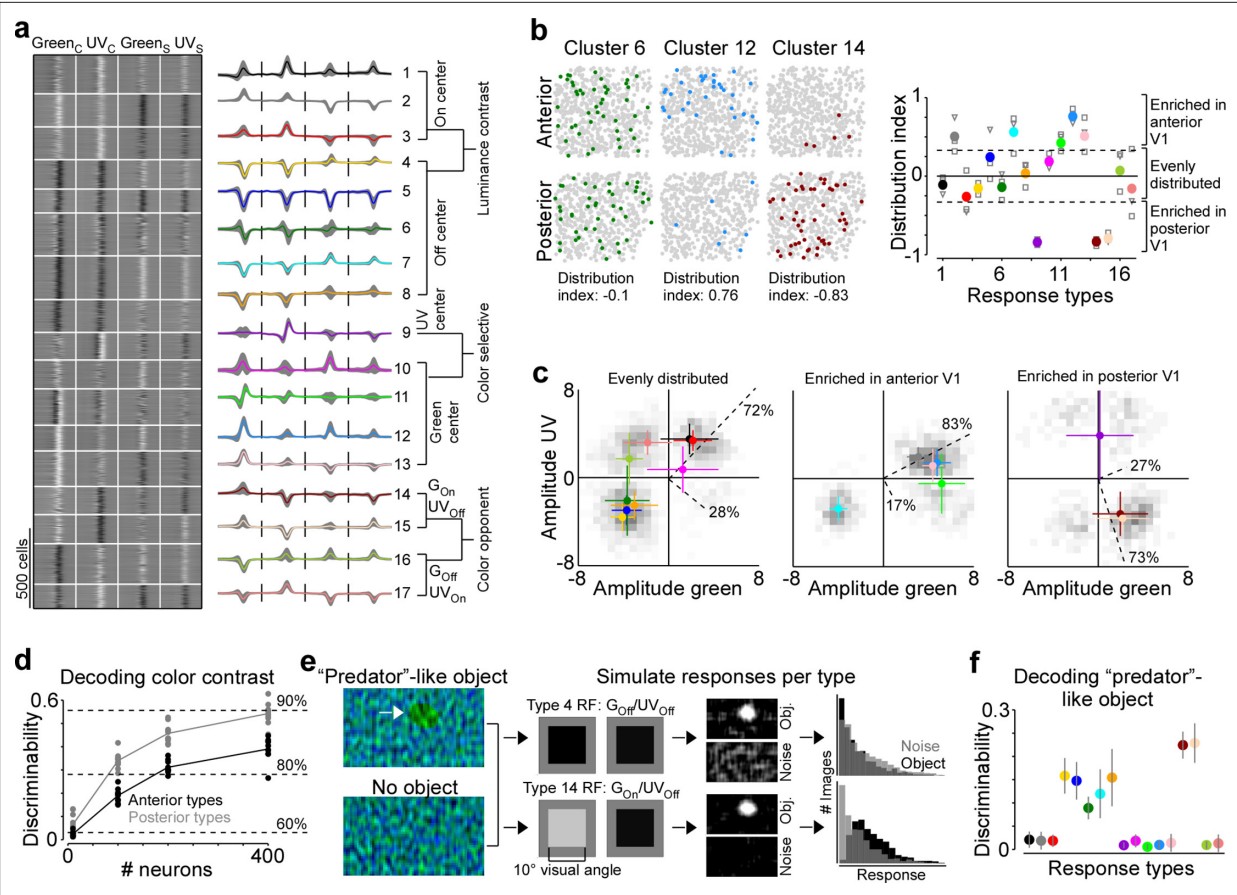

**Figure 5.** Asymmetric distribution of color response types explains higher color sensitivity in posterior primary visual cortex (V1). (**a**) Clustering result of the Gaussian mixture model with *n*=17 clusters (see **Figure 5—figure supplement 1** for details). Model input corresponded to the weights of the principal components used for reconstructing event-triggered averages (ETAs). Left panel shows ETAs of all cells, respectively, sorted by cluster assignment. Right panel shows mean ETA of each cluster (s.d. shading in gray). Clusters are sorted based on broad response categories, which are indicated on the right. (**b**) Left shows distribution of cells assigned to three different clusters (color) in a posterior and anterior recording field of one example animal. Gray dots show cells assigned to other clusters. Distribution index for each cluster is indicated below. Right shows the mean distribution index per cluster, with different marker shapes indicating the indices for individual animals. Zero indicates an even distribution across anterior and posterior V1 and values above and below zero indicate that cells are enriched in anterior and posterior V1, respectively. Dotted horizontal lines at –0.33/0.33 indicate twice as many cells in posterior than anterior cortex and vice versa. (**c**) Histograms of peak amplitudes of center ETAs for clusters that are evenly distributed across V1 (left, distribution index >–0.33 and <0.33), enriched in anterior V1 (middle, distribution index) and enriched in posterior V1 (right). Cluster means and s.d. are indicated in color. Dotted lines correspond to axes of PCs obtained from a principal component analysis (PCA), with percentage of variance explained along the luminance and color axis indicated. The percentages of variance explained by color (off-diagonal) and luminance axis (diagonal) correlate with the number of neurons located in the color (top left and bottom right) and luminance contrast quadrants (top right and bottom left), respectively. Scale bars indicate the number of neurons in the 2D histogram. (**d**) Discriminability (in bits) of stimulus color contrast based on response types enriched in anterior (black) and posterior (gray) V1. Dots show decoding performance across 10 train/test trial splits. Decoding discriminability was significantly different between anterior- and posterior-enriched types for all numbers of neurons tested (t-test for unpaired data, p-value was adjusted for multiple comparisons using Bonferroni correction). Dotted horizontal lines indicate decoding accuracy in % for 60%, 80%, and 90%, with a change level of 50% corresponding to 0 bits. (**e**) Noise images with or without a 'predator'-like dark object in the UV channel were convolved with simulated center receptive fields (RFs), depicting the mean amplitudes of the green and UV center ETA per response type (shown for types 4 and 14). The resulting activity maps were summed and thresholded to simulate responses to *n*=1000 noise and object scenes. (**f**) Discriminability (in bits) of the presence of a 'predator'-like dark object in the UV channel per response type. Error bars show s.d. across 10 train/test trial splits.

The online version of this article includes the following figure supplement(s) for figure 5:

**Figure supplement 1.** Unsupervised clustering of spike-triggered averages.

the green wavelength range. For each response type, we first simulated responses to these scenes based on the type's luminance and color contrast sensitivity of the RF center using a simple linear-nonlinear model (*Figure 5e*). We found that all UV-Off types responded to the predator-like object, but the green-On/UV-Off response types did so more selectively, without also responding to the noise scenes. This selectivity arises because driving color-opponent neurons requires the mean intensity of the UV and green channels within the RF to have opposite polarity, a condition less likely to occur in noisy scenes compared to when the mean intensity is lower than the background. We then used the simulated responses to train an SVM decoder to discriminate between object and noise-only scenes. While all Off-center response types sensitive to luminance contrast could decode the dark object, the two best performing types corresponded to the green-On/UV-Off response types enriched in posterior V1 (*Figure 5f*). Interestingly, the reason for their good performance was the absence of responses to the noise scenes, rather than strong responses to the object scenes per se (*Figure 5e*). Our results suggest that functional neuron types in mouse V1 with distinct color properties unevenly sample different parts of the visual scene, and might thereby serve a distinct role in driving visually guided behavior like predator detection.

## Discussion

Here, we found that a large fraction of neurons in mouse visual cortex encode color features of the visual input in their RF center. Color-opponency was strongest for photopic light levels and especially pronounced in posterior V1 encoding the sky. This asymmetry in color processing across visual space was due to an inhomogeneous distribution of color-opponent response types, with green-On/UV-Off response types predominantly being present in posterior V1. Using a simple model and natural scene inspired parametric stimuli, we showed that this type of color-opponency may enhance the detection of aerial predators in noisy daylight scenes.

### Neuronal correlates of color vision in mice

In most species, color vision is mediated by comparing signals from different cone photoreceptor types sensitive to different wavelengths (reviewed in *Baden and Osorio, 2019*). This includes circuits with cone-type selective wiring present in many vertebrate species and circuits with random and cone type-unselective wiring like red-green opponency in primates. Recently, it has been demonstrated that there is extensive rod-cone opponency in mice, comparing signals from UV-sensitive cones in the ventral retina to rod signals (*Szatko et al., 2020*; *Joesch and Meister, 2016*; *Rhim and Nauhaus, 2023*; *Khani and Gollisch, 2021*). In the retina, this mechanism relies on integrating information from cone signals in the RF center with rod signals in the RF surround (*Szatko et al., 2020*; *Khani and Gollisch, 2021*; *Joesch and Meister, 2016*). Interestingly, there is also evidence for rod-cone opponency in monochromatic humans (*Reitner et al., 1991*), suggesting that a neuronal circuit to compare rod and cone signals exists in other mammals as well. At this point, it is still unclear to what extent behavioral color discrimination in mice (*Denman et al., 2018*; *Jacobs et al., 2004*) is driven by rod-cone versus cone-cone comparisons.

Here, we found that the neuronal representation of color in mouse visual cortex is most prominent for photopic light levels and decreases for mesopic conditions, indicating that color-opponency in mouse V1 is at least partially mediated by the comparison of cone signals and not purely by cone-rod comparisons. Our result is consistent with a recent study reporting pronounced cone-mediated color-opponency in mouse dLGN (*Mouland et al., 2021*). In our experimental paradigm, dissecting the relative contribution of rods and M-cones in color-opponency of mouse V1 neurons was not possible, due to the highly overlapping wavelength sensitivity profiles of mouse M-opsin and Rhodopsin. However, it is very likely that rods contribute to the prominent color-opponent neuronal responses we observed in mouse V1. This is especially true for posterior V1 receiving input from the ventral retina where cone-cone comparisons are challenging due to the co-expression of S-opsin in M-cones (*Szél et al., 1992*; *Baden et al., 2013*). The involvement of rods in generating color-opponent responses is supported by retinal data (*Szatko et al., 2020*; *Khani and Gollisch, 2021*; *Joesch and Meister, 2016*) and a recent study performed in mouse visual cortex showing that color-opponency in posterior V1 is best explained by a model that compares S-opsin with Rhodopsin (*Rhim and Nauhaus, 2023*). Surprisingly, we found that the mouse visual system still extracts color information for relatively low light levels

present during early dusk and dawn, potentially by comparing rod to remaining cone signals. Future studies will tell whether color-opponency under dim light, as observed here for low mesopic conditions, require a specific neuronal pathway amplifying the relatively weak cone signals to encode color features present in the environment.

The results from our study, together with recent findings across the visual hierarchy of mice (*Mouland et al., 2021*; *Rhim and Nauhaus, 2023*; *Szatko et al., 2020*; *Khani and Gollisch, 2021*), demonstrate a pronounced neuronal representation of color in mouse visual brain areas that is mediated by both cone-cone and cone-rod comparisons. While this highlights the relevance of color information in mouse vision, it remains unclear as to how mice use color vision to inform natural behaviors. Two behavioral studies using parametric stimuli and relatively simple behavioral paradigms have shown that mice can discriminate different colors (*Jacobs et al., 2004*; *Denman et al., 2018*), at least in their central and upper visual field (*Denman et al., 2018*). Here, we found that green-Off/UV-On color-opponency was equally distributed across cortex, suggesting that there is a neuronal substrate for color vision in mice across the entire visual field. In contrast, green-On/UV-Off color-opponency was confined to posterior V1, where it might aid the detection of aerial predators present in cluttered and noisy daylight scenes, such as in the forest. Testing this hypothesis and further elucidating the role of color vision in mouse natural behaviors will require combining more unrestrained behavioral paradigms with ecologically relevant stimuli.

## Limitations of the stimulus and analysis paradigm

To study color processing in the mouse V1, we employed a parametric center-surround color flicker stimulus, similar to those used in previous studies (*Chatterjee and Callaway, 2003*; *Zimmermann et al., 2018*; *Szatko et al., 2020*). The advantage of this relatively simple approach is that it allows for clear interpretation of neuronal responses using linear methods, such as the spike-triggered average method (*Schwartz et al., 2006*). However, a limitation of the linear ETA approach employed here is its potential inadequacy in capturing the stimulus selectivity of nonlinear neurons. Our linear analysis provides preliminary insights into color representations in the mouse visual cortex, which future studies could enhance with nonlinear methods to achieve a more comprehensive understanding. Additionally, the ETA method may be less effective for neurons that respond to both On and Off stimuli with similar amplitudes. Although we cannot rule out that our analysis may have been biased against On-Off neurons, the observation that over 62% of neurons exhibited a significant ETA for their RF center suggests that we captured a substantial proportion of V1 neurons.

For an ETA analysis, the stimulus should ideally be aligned to the center RF of each neuron, which requires detailed RF mapping of individual neurons. As this procedure is relatively time consuming and low throughput, we instead used a center stimulus that was slightly larger than RFs of single neurons, and was centered on the mean RF across a population of V1 neurons. To reduce the impact of potential stimulus misalignment on the single cell level on our results, we used different experimental steps and controls, such as confirming that the RF center of most recorded neurons greatly overlaps with the center noise stimulus. The fact that response types identified using an automated clustering approach were consistent across animals suggests that stimulus alignment did not significantly contribute to the neurons' visual responses. Nevertheless, we cannot exclude that the stimulus was misaligned for a subset of the recorded neurons used for analysis. Stimulus misalignment might have contributed to single cells not having surround ETAs, due to simultaneous activation of antagonistic center and surround RF components by the surround stimulus.

## Asymmetric processing of color information across the visual field

The spatial arrangements of sensory neurons are ordered in a way that encodes particular characteristics of the surrounding environment. One classical example in the visual system is that the density of all retinal output neurons increases and their dendritic arbor size decreases toward retinal locations with higher sampling frequency, such as the fovea in primates and the area centralis in carnivores (discussed in *Peichl, 2005*). More recent research has uncovered how the visual circuits in certain species are customized to suit the statistics of the visual information they receive, including the distribution of spatial, temporal, and spectral information, as well as the specific requirements of their behavior (discussed in *Baden et al., 2020*). For example, a study in zebrafish larvae showed that UV

cones in one particular retinal location are specifically tuned for UV-bright objects, thereby supporting prey capture in their upper frontal visual field (*Yoshimatsu et al., 2020*).

Here, we found that there is a pronounced asymmetry in how color is represented across visual space in mouse V1. A similar asymmetry in color processing was reported at the level of the mouse retina (*Szatko et al., 2020*; *Khani and Gollisch, 2021*) and dLGN (*Mouland et al., 2021*), and has been linked to an inhomogeneous distribution of color contrast across natural scenes from the mouse's environment (*Qiu et al., 2021*; *Abballe and Asari, 2022*). Specifically, it has been speculated that the higher color contrast present in the upper visual field of natural scenes captured in the natural habitat of mice might have driven superior color-opponency in the ventral retina (*Qiu et al., 2021*), thereby supporting color discrimination in the sky (*Denman et al., 2018*). Our results extend these previous studies by demonstrating that the asymmetry across visual cortex can be explained by the asymmetric distribution of response types with distinct color tuning in their RF center, and by linking them to a neuronal computation relevant for the upper visual field, namely the detection of aerial predators.

At the level of the mouse retina, color-opponency is largely mediated by center-surround interactions (*Szatko et al., 2020*; *Khani and Gollisch, 2021*; *Joesch and Meister, 2016*), with only a few neurons exhibiting color-opponency in their centers (*Hoefling et al., 2022*). Consistent with this, we found that the pronounced representation of color by the center component of V1 RFs was not solely inherited from the color-opponency present in the RF centers of retinal output neurons. Similarly, a recent study concluded that the extensive and sophisticated color processing in the mouse LGN cannot be fully explained by the proposed retinal opponency mechanisms (*Mouland et al., 2021*). However, we compared properties of the center components of retinal (3–10 degrees visual angle) and V1 RFs (10–25 degrees visual angle), which differ in size. Therefore, the integration of center and surround retinal signals might still contribute to the color-opponency observed in downstream visual areas, such as the mouse V1, as observed in this study. Generally, comparing ex vivo retinal data with in vivo cortical data is challenging, not only due to differences in RF size but also due to varying levels of adaptation.

## Strategies of color processing across animal species: distributed versus specialized code

In primates, physiological and anatomical evidence suggest that a small number of distinct retinal cell types transmit color information to downstream visual areas (reviewed in *Thoreson and Dacey, 2019*), where the neuronal representation of color remains partially segregated from the representation of other visual features like form (*Livingstone and Hubel, 1988*; *Zeki, 1978*, but see *Garg et al., 2019*). For example, color-sensitive neurons in primary and secondary visual cortex are enriched in the so-called 'blob' (*Hubel and Livingstone, 1987*) and 'inter-stripe' regions (*DeYoe and Van Essen, 1985*), respectively. Interestingly, in other vertebrate species, color processing is distributed across many neuron types and cannot easily be separated from the processing of other visual features. In zebrafish, birds, *Drosophila*, and mice, a large number of retinal output types encode information about stimulus color (*Seifert et al., 2023*; *Zhou et al., 2020*; *Szatko et al., 2020*; *Khani and Gollisch, 2021*), in addition to each type's preferred feature like direction of motion. In addition, there is evidence for distributed processing of color in visual areas downstream to the retina in zebrafish (*Guggiana Nilo et al., 2021*), mice (*Rhim and Nauhaus, 2023*; *Mouland et al., 2021*), *Drosophila* (*Longden et al., 2023*), and tree shrew (*Johnson et al., 2010*). Our results demonstrate a prominent neuronal representation of color in mouse V1, which is distributed across many neurons and multiple response types.

What might be the benefit of such a distributed code of color processing? It is important to note that chromatic signals may not only be used for color discrimination per se, but instead different spectral channels might facilitate the extraction of specific features from the environment. For example, it has been shown that the UV wavelength range aids the detection of objects like prey, predators, and food (reviewed in *Cronin and Bok, 2016*) by increasing their contrast, as recently shown for leaf surface contrasts in forest environments (*Tedore and Nilsson, 2019*). Indeed, it is hypothesized that different photoreceptor types sensitive to distinct wavelength bands did not evolve to support color discrimination, but instead to reduce lighting noise in the natural environment of early vertebrates (discussed in *Maximov, 2000*; *Kelber et al., 2003*). In line with this idea, our analysis suggests that green-On/UV-Off color-opponency might facilitate the detection of predatory-like dark objects in the

UV channel by reducing the neurons' activation to noise, rather than increasing the neurons' activation to the object. If chromatic signals are predominantly used to boost contrast of specific aspects of the environment, it might make sense to widely distribute chromatic tuning and color-opponency across visual neurons. Further experiments and analysis will uncover the computational relevance of the pronounced and distributed color representations observed in mice and other vertebrate species.

## Materials and methods

### Neurophysiological experiments

All procedures were approved by the Institutional Animal Care and Use Committee of Baylor College of Medicine (animal protocol number: AN-4703). Owing to the explanatory nature of our study, we did not use randomization and blinding. No statistical methods were used to predetermine sample size.

Mice of either sex (*Mus musculus*, $n$=9; 2–5 months of age) expressing GCaMP6s in excitatory neurons via Slc17a7-Cre and Ai162 transgenic lines (stock number 023527 and 031562, respectively; The Jackson Laboratory) were anesthetized and a 4 mm craniotomy was made over the visual cortex of the right hemisphere as described previously (*Reimer et al., 2014*; *Froudarakis et al., 2014*). For functional recordings, awake mice were head-mounted above a cylindrical treadmill and calcium imaging was performed using a Ti-Sapphire laser tuned to 920 nm and a two-photon microscope equipped with resonant scanners (Thorlabs) and a ×25 objective (MRD77220, Nikon). Laser power after the objective was kept below 60 mW. The rostro-caudal treadmill movement was measured using a rotary optical encoder with a resolution of 8000 pulses per revolution. We used light diffusing from the laser through the pupil to capture eye movements and pupil size. Images of the pupil were reflected through a hot mirror and captured with a GigE CMOS camera (Genie Nano C1920M; Teledyne Dalsa) at 20 fps at a 1920×1200 pixel resolution. The contour of the pupil for each frame was extracted using DeepLabCut (*Mathis et al., 2018*) and the center and major radius of a fitted ellipse were used as the position and dilation of the pupil.

For image acquisition, we used ScanImage. To identify V1 boundaries, we used pixelwise responses to drifting bar stimuli of a 2400×2400 $\mu m^2$ scan at 200 µm depth from cortical surface (*Garrett et al., 2014*), recorded using a large field of view mesoscope (*Sofroniew et al., 2016*). Functional imaging was performed using 512×512 pixel scans (700×700 $\mu m^2$) recorded at approx. 15 Hz and positioned within L2/3 (depth 200 µm) in posterior or anterior V1. Imaging data were motion-corrected, automatically segmented, and deconvolved using the CNMF algorithm (*Pnevmatikakis et al., 2016*); cells were further selected by a classifier trained to detect somata based on the segmented masks. This resulted in approx. 500–1200 selected soma masks per scan depending on response quality and blood vessel pattern.

To achieve photopic stimulation of the mouse visual system, we dilated the pupil pharmacologically with atropine eye drops (*Franke et al., 2022*). Specifically, atropine was applied to the left eye of the animal facing the screen for visual stimulation. Functional recordings started after the pupil was dilated. Pharmacological pupil dilation lasted >2 hr, thereby ensuring a constant pupil size during all functional recordings.

### Visual stimulation

Visual stimuli were presented to the left eye of the mouse on a 42×26 $cm^2$ light-transmitting Teflon screen (McMaster-Carr) positioned 12 cm from the animal, covering approx. 120×90 degrees visual angle. Light was back-projected onto the screen by a DLP-based projector (EKB Technologies Ltd; *Franke et al., 2019*) with UV (395 nm) and green (460 nm) LEDs that differentially activated mouse S- and M-opsin. LEDs were synchronized with the microscope's scan retrace.

Light intensity (estimated as photoisomerization rate, P* per second per cone) was calibrated using a spectrometer (USB2000+, Ocean Optics) to result in equal activation rates for mouse M- and S-opsin (for details, see *Franke et al., 2019*). In brief, the spectrometer output was divided by the integration time to obtain counts/s and then converted into electrical power (in nW) using the calibration data (in µJ/count) provided by Ocean Optics. To obtain the estimated photoisomerization rate per photoreceptor type, we first converted electrical power into energy flux (in eV/s) and then calculated the photon flux (in photons/s) using the photon energy (in eV). The photon flux density (in photons/s/$\mu m^2$) was then computed and converted into photoisomerization rate using the effective activation

of mouse cone photoreceptors by the LEDs and the light collection area of cone outer segments. In addition, we considered both the wavelength-specific transmission of the mouse optical apparatus (*Henriksson et al., 2010*) and the ratio between pupil size and retinal area (*Schmucker and Schaeffel, 2004*). Please see the calibration iPython notebook provided online for further details.

We used three different light levels, ranging from photopic levels primarily activating cone photo-receptors to low mesopic levels that predominantly drive rod photoreceptors. For a mean pupil size across recordings within one light level and a maximal stimulus intensity (255 pixel values), this resulted in 50 P*, 400 P*, and 15,000 P* for low mesopic, high mesopic, and photopic light levels, respectively. Please note that the difference between photopic and high mesopic light levels is higher than between high and low photopic light levels.

Prior to functional recordings, the screen was positioned such that the population RF across all neurons, estimated using an achromatic sparse noise paradigm, was within the center of the screen. Screen position was fixed and kept constant across recordings of the same neurons. We used Psych-toolbox in MATLAB for stimulus presentation and showed the following light stimuli.

## Center-surround luminance and color noise

We used a center (diameter: 37.5 degrees visual angle) and surround (full screen except the center) binary noise stimulus of UV and green LED to characterize center and surround chromatic properties of mouse V1 neurons. For that, the intensity of UV and green center and surround spots was determined independently by a binary and balanced 25 min random sequence updated at 5 Hz. A similar stimulus was recently used in recordings of the mouse retina (*Szatko et al., 2020*). The center size of 37.5 degrees visual angle in diameter is larger than the mean center RF size of mouse V1 neurons (26.2±4.6 degrees visual angle in diameter). This allowed to record from a large neuron population, despite some variability in RF center location. We verified that the center RF of the majority of neurons lies within the center spot of the noise stimulus using a sparse noise stimulus for spatial RF mapping (*Figure 1—figure supplement 1a, b*).

## Sparse noise

To map the spatial RFs of V1 neurons, we used a sparse noise paradigm. UV and green bright (pixel value 255) and dark (pixel value 0) dots of approx. 12 degrees visual angle were presented on a gray background (pixel value 127) in randomized order. Dots were presented for 8 and 5 positions along the horizontal and vertical axis of the screen, respectively, excluding screen margins. Each presentation lasted 200 ms and each condition (e.g. UV-bright dot at position *x*=1 and *y*=1) was repeated 50 times.

## Preprocessing of neural responses and behavioral data

Neuronal calcium responses were deconvolved using constrained non-negative calcium deconvolution (*Pnevmatikakis et al., 2016*) to obtain estimated spike trains. For the decoding paradigm, we subsequently extracted the accumulated activity of each neuron between 50 ms after stimulus onset and offset using a Hamming window. Behavioral traces (treadmill velocity and pupil size) were synchronized to the recorded neuronal response traces, but not used for further processing - i.e., we did not distinguish between arousal states of the animal.

## RF mapping based on the center-surround color noise stimulus

We used the responses to the 5 Hz center-surround noise stimulus of UV and green LED to compute temporal ETAs of V1 neurons. Specifically, we upsampled both stimulus and responses to 30 Hz, normalized each upsampled response trace by its sum and then multiplied the stimulus matrix with the response matrix for each neuron. Per cell, this resulted in a temporal ETA for center (C) and surround (S) in response to UV and green flicker, respectively (Green$_C$, UV$_C$, Green$_S$, UV$_C$). For each of the four stimulus conditions, kernel quality was measured by comparing the variance of the ETA with the variance of the baseline, defined as the first 500 ms of the ETA. Only cells with at least 10 times more variance of the kernel compared to baseline for UV or green center ETA were considered for further analysis.

## Sparse noise spatial RF mapping and overlap index

We estimated spatial ETAs of V1 neurons in response to the sparse noise stimulus by multiplying the stimulus matrix with the response matrix of each neuron (*Schwartz et al., 2006*). For that, we

averaged across On and Off and UV and green stimuli, thereby obtaining a two-dimensional (8×5 pixels) spatial ETA per neuron. To assess ETA quality, we generated response predictions by multiplying the flattened ETA of each neuron with the flattened stimulus frames and compared the predictions to the recorded responses by estimating the linear correlation coefficient. For analysis, we only included cells where correlation >0.25. For these cells, we upsampled and peak-normalized the spatial ETAs (resulting in 40×25 pixels), and then estimated the overlap with the center spot of the noise stimulus using a contour threshold of 0.25. Specifically, we calculated the ratio of pixels >0.25 with respect to the peak of the ETA inside and outside the area of the noise center spot.

## PCA for ETA reconstruction

To increase the signal-to-noise ratio of the ETAs, we converted them into lower dimensional representations using sparse PCA (*Figure 1—figure supplement 3*). Specifically, we concatenated the ETAs of the four stimulus conditions and used the resulting matrix with dimensions *neurons × time* to perform sparse PCA using the package *sklearn.decomposition.SparsePCA* in Python. We used sparse PCA because each principal component (PC) then captured one of the four stimulus conditions (*Figure 1—figure supplement 3b*), thereby making the PCs interpretable. We tested different numbers of PCs (*n*=2 to *n*=12 components) and evaluated the quality of the PCA reconstructions by computing the mean squared error (*mse*) between the original ETA and the one reconstructed based on the PCs. We decided to use eight PCs because (i) reconstruction *mse* dropped only slightly with more PCs (*Figure 1—figure supplement 3a*) and (ii) additional PCs captured variance outside the time window of expected stimulus sensitivity, e.g., after the response time.

## Spectral contrast

For estimating the chromatic preference of the recorded neurons, we used spectral contrast (*SC*). It is estimated as Michelson contrast ranging from –1 to 1 for a neuron responding solely to UV and green contrast, respectively. We define *SC* as

$$SC = \frac{r_{green} - r_{UV}}{r_{green} + r_{UV}}$$

where $r_{green}$ and $r_{UV}$ correspond to the amplitude of UV and green ETA to estimate the neurons' chromatic preference.

## Luminance and color contrast sensitivity space

To represent each neuron in a two-dimensional luminance and color contrast space, we extracted ETA peak amplitudes relative to baseline for all four stimulus conditions, with positive and negative peak amplitudes for On and Off cells, respectively. Peak amplitudes of green and UV ETA were then used as *x* and *y* coordinates, respectively, in the two-dimensional contrast spaces for center and surround. To obtain the fraction of variance explained by the luminance and color axis within the contrast space for center and surround RF components, we performed PCA on the two-dimensional matrix with dimensions *cells × x − y*. The relative weights of the resulting PCs were used as a measure of fraction variance explained. A similar method was recently used to quantify chromatic and achromatic contrasts in mouse natural scenes (*Qiu et al., 2021*).

## Decoding analysis

We used an SVM classifier with a radial basis function kernel to estimate decoding accuracy between the neuronal representations of two stimulus classes - either On or Off (stimulus luminance) and UV or green (stimulus color). We used varying numbers of neurons for decoding and built separate decoders for stimulus luminance and stimulus color. Specifically, we split the data into 10 equally sized trial blocks, trained the decoder on 90% of the data, tested its accuracy on the remaining 10% of the data, and computed the mean accuracy across *n*=10 different training/test trial splits. Finally, we converted the decoding accuracy into discriminability, the mutual information between the true class and its estimate using

$$MI\left(c, \hat{c}\right) = \sum_i \sum_j p_{ij} \log_2 \frac{p_{ij}}{p_{i:} \cdot p_{:j}}$$

where $p_{ij}$ is the probability of observing the true class $i$ and predicted class $j$ and $p_{i:}$ and $p_{:j}$ denote the respective marginal probabilities.

## Retinal data

We used an available dataset from *Szatko et al., 2020*, to test how color is represented in the luminance and color contrast space at the level of the retinal output. This dataset consisted of UV and green center and surround ETAs of $n$=3215 retinal ganglion cells ($n$=88 recording fields, $n$=18 mice), obtained from responses to a center (10 degrees visual angle) and surround (30×30 degrees visual angle without the center) luminance and color noise stimulus. We estimated the ETAs and embedded each neuron in the sensitivity space as described above.

## Functional clustering using GMM

For clustering of center and surround ETAs into distinct response types, we used a GMM (*sklearn.mixture.GaussianMixture* package). We used the weights of the PCs extracted from the ETAs as input to the GMM (*Figure 1—figure supplement 3*). To test how many GMM components (i.e. response types) best explain the data, we built GMMs with varying numbers of components and cross-validated the models' log likelihood on 10% of left-out test data, using 10 different test/train trial splits (*Figure 5—figure supplement 1a*). We picked the model with $n$=17 components for further analysis because this resulted in the highest log likelihood. However, please note that the models' performance was relatively ETAble across a wide range of components. To test the assignment accuracy of the final model, we used the mean and covariance matrix of each GMM component to generate data with ground-truth labels and compared those to the GMM-predicted labels (*Figure 5—figure supplement 1b*), as described previously (*Tolias et al., 2007*). Assignment accuracy ranged between 75% and 98%, with a mean ± s.d. of 89% ± 6%. Most response types were evenly distributed across mice and all response types were present in all mice (*Figure 5—figure supplement 1c*), suggesting that clustering was not predominantly driven by inter-experimental variations.

## Cortical distribution index

For estimating the distribution of response types across cortical position, we used the cortical distribution index. It was estimated as Michelson contrast ranging from –1 to 1 for a response type solely present in posterior and anterior V1, respectively. We define the distribution index as

$$Distribution_{Index} = \frac{n_{anterior} - n_{posterior}}{n_{anterior} + n_{posterior}}$$

where $n_{anterior}$ and $n_{posterior}$ correspond to the fraction of neurons in anterior and posterior V1 assigned to a specific response type.

## Decoding of noise and object scenes

For decoding noise versus object scenes based on simulated responses, we used natural scene inspired parametric stimuli. Specifically, we generated images with independent Perlin noise *Perlin, 1985* in each color channel using the Perlin-noise package for Python. Then, for the object images, we added a dark ellipse of varying size, position, and angle to the UV color channels. We adjusted the contrast of all images with a dark object to match the contrast of noise images, such that the distribution of image contrasts did not differ between noise and object images. We then simulated responses to 1000 object and noise scenes that were used by an SVM decoder to decode stimulus class (object or noise) as described above. For simulating responses, we modeled each response type to have a square RF with 10 degrees visual angle in diameter, with the luminance and color contrast sensitivity of the response type's RF center. Then, we created response maps by convolving the simulated RFs with the scenes and summed up all positive values to result in one response value per scene and response type.

## Statistical analysis

We used the t-test for two independent samples to test whether the decoding performance of 10 test/train trial splits differ between (i) center and surround, (ii) photopic and mesopic light levels,

(iii) anterior and posterior V1, and (iv) anterior and posterior response types. For all these tests, the p-value was adjusted for multiple comparisons using the Bonferroni correction.

## Acknowledgements

We thank Thomas Euler and Tom Baden for feedback on the manuscript. This work was supported by the Hertie Foundation (to PB), the German Research Foundation (DFG CRC 1233 'Robust Vision' to PB and KF, DFG Excellence Cluster 2064 'Machine Learning - New Perspectives for Science' to PB), the European Research Council (ERC) under the European Union's Horizon Europe research and innovation programme ('NextMechMod' grant agreement No. 101039115 to PB), and the National Institutes of Health (UF1 NS126566 and RF1 MH126883 to AST).

## Additional information

### Funding

| Funder | Grant reference number | Author |
| --- | --- | --- |
| Hertie-Stiftung | | Philipp Berens |
| Deutsche Forschungsgemeinschaft | DFG CRC 1233 | Katrin Franke Philipp Berens |
| Deutsche Forschungsgemeinschaft | DFG Excellence Cluster 2064 | Philipp Berens |
| European Research Council | 101039115 | Philipp Berens |
| National Institutes of Health | UF1 NS126566 | Andreas Savas Tolias |
| National Institutes of Health | RF1 MH126883 | Andreas Savas Tolias |

The funders had no role in study design, data collection and interpretation, or the decision to submit the work for publication.

### Author contributions

Katrin Franke, Conceptualization, Software, Formal analysis, Supervision, Funding acquisition, Validation, Investigation, Visualization, Methodology, Writing - original draft, Project administration; Chenchen Cai, Formal analysis, Validation, Visualization, Writing - review and editing; Kayla Ponder, Investigation, Data acquisition; Jiakun Fu, Investigation, Methodology, Writing - review and editing; Sacha Sokoloski, Formal analysis, Methodology, Writing - review and editing; Philipp Berens, Supervision, Funding acquisition, Methodology, Writing - review and editing; Andreas Savas Tolias, Conceptualization, Supervision, Funding acquisition, Investigation, Writing - review and editing

### Author ORCIDs

Katrin Franke ⓘ https://orcid.org/0000-0002-8649-4835
Chenchen Cai ⓘ https://orcid.org/0009-0007-3724-5203
Sacha Sokoloski ⓘ https://orcid.org/0000-0003-4166-1772
Philipp Berens ⓘ https://orcid.org/0000-0002-0199-4727

### Ethics

All procedures were approved by the Institutional Animal Care and Use Committee of Baylor College of Medicine (animal protocol number: AN-4703).

Reviewer #1 (Public Review): https://doi.org/10.7554/eLife.89996.4.sa1
Reviewer #2 (Public Review): https://doi.org/10.7554/eLife.89996.4.sa2
Reviewer #3 (Public Review): https://doi.org/10.7554/eLife.89996.4.sa3
Author response https://doi.org/10.7554/eLife.89996.4.sa4

## Additional files

**Supplementary files**
- MDAR checklist

**Data availability**
The data and code is available at: https://doi.org/10.5061/dryad.rv15dv4hb.

The following dataset was generated:

| Author(s) | Year | Dataset title | Dataset URL | Database and Identifier |
|-----------|------|---------------|-------------|-------------------------|
| Franke K | 2024 | Data from: Asymmetric distribution of color-opponent response types across mouse visual cortex supports superior color vision in the sky | http://dx.doi.org/10.5061/dryad.rv15dv4hb | Dryad Digital Repository, 10.5061/dryad.rv15dv4hb |

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
