## [Editor Report · eLife assessment]

Franke et al. explore and characterize color response properties of neurons in mouse primary visual cortex (V1), revealing specific color opponent encoding strategies across the visual field. The paper provides evidence for the existence of color opponency in a subset of neurons within V1 and shows that these color opponent neurons are more numerous in the upper visual field. Support for the main conclusions is **convincing** and the dataset that forms the basis of the paper is impressive. The paper will make an **important** contribution to understanding how color is coded in mouse V1.

---

## [Referee Report · Reviewer #1 (Public Review)]

In this study, Franke et al. explore and characterize color response properties across primary visual cortex, revealing specific color opponent encoding strategies across the visual field. The authors use awake 2P imaging to define the spectral response properties of visual interneurons in layer 2/3. They find that opponent responses are more pronounced at photopic light levels, and that diversity in color opponent responses exists across the visual field, with green ON/ UV OFF responses more strongly represented in the upper visual field. This is argued to be relevant for the detection of certain features that are more salient when using chromatic space, possibly due to noise reduction. In the revised version, Franke et al. have addressed the potential pitfalls in the discussion, which is an important point for the non-expert reader. Thus, this study provides a solid characterization of the color properties of V1 and is a valuable addition to visual neuroscience research.

---

## [Referee Report · Reviewer #2 (Public Review)]

Summary:

Franke et al. characterize the representation of color in the primary visual cortex of mice, highlighting how this changes across the visual field. Using calcium imaging in awake, head-fixed mice, they characterize the properties of V1 neurons (layer 2/3) using a large center-surround stimulation where green and ultra-violet colors were presented in random combinations. Clustering of responses revealed a set of functional cell-types based on their preference to different combinations of green and UV in their center and surround. These functional types were demonstrated to have different spatial distributions across V1, including one neuronal type (Green-ON/UV-OFF) that was much more prominent in the posterior V1 (i.e. upper visual field). Modelling work suggests that these neurons likely support the detection of predator-like objects in the sky.

Strengths:

The large-scale single-cell resolution imaging used in this work allows the authors to map the responses of individual neurons across large regions of the visual cortex. Combining this large dataset with clustering analysis enabled the authors to group V1 neurons into distinct functional cell types and demonstrate their relative distribution in the upper and lower visual fields. Modelling work demonstrated the different capacity of each functional type to detect objects in the sky, providing insight into the ethological relevance of color opponent neurons in V1.

Weaknesses:

It is unfortunate the authors were unable to provide stronger mechanistic insights into how color opponent neurons in V1 are formed.

Overall, this study will be a valuable resource for researchers studying color vision, cortical processing, and the processing of ethologically relevant information. It provides a useful basis for future work on the origin of color opponency in V1 and its ethological relevance.

---

## [Referee Report · Reviewer #3 (Public Review)]

This paper improves our understanding of the coding of chromatic signals in mouse visual cortex. Calcium responses of a large collection of cells are measured in response to a simple spot stimulus. These responses are used to estimate chromatic tuning properties - specifically sensitivity to UV and green stimuli presented in a large central spot or a larger still surrounding region. Cells are divided based on their responses to these stimuli into luminance or chromatic sensitive groups.

The paper has improved substantially in revisions and makes an important contribution to how color is coded in mouse V1. The revisions have nicely clarified a few limitations of the current study, and that serves to emphasize the strengths of the data and clear conclusions that can be drawn from it.

---

## [Author Response]

The following is the authors’ response to the previous reviews.

**eLife assessment:**
Franke et al. explore and characterize the color response properties in the mouse primary visual cortex, revealing specific color opponent encoding strategies across the visual field. The data is solid; however, the evidence supporting some conclusions is incomplete. In its current form, the paper makes a useful contribution to how color is coded in mouse V1. Significance would be enhanced with some additional analyses and a clearer discussion of the limitations of the data presented.

We thank the reviewers for appreciating our manuscript. We have rewritten the conclusions of the paper to be more conservative and now more explicitly focus on color processing in mouse V1, rather than comparing V1 to the retina. Additionally, we discuss the limitations of our approach in detail in the Discussion section. Finally, we have addressed all comments from the reviewers below.

**Referee 1 (Remarks to the Author):**
In this study, Franke et al. explore and characterize color response properties across primary visual cortex, revealing specific color opponent encoding strategies across the visual field. The authors use awake 2P imaging to define the spectral response properties of visual interneurons in layer 2/3. They find that opponent responses are more pronounced at photopic light levels, and that diversity in color opponent responses exists across the visual field, with green ON/ UV OFF responses more strongly represented in the upper visual field. This is argued to be relevant for the detection of certain features that are more salient when using chromatic space, possibly due to noise reduction. In the revised version, Franke et al. have addressed the potential pitfalls in the discussion, which is an important point for the non-expert reader. Thus, this study provides a solid characterization of the color properties of V1 and is a valuable addition to visual neuroscience research.My remaining concerns are based more on the interpretation. I’m still not convinced by the statement "This type of color-opponency in the receptive field center of V1 neurons was not present in the receptive field center of retinal ganglion cells and, therefore, is likely computed by integrating center and surround information downstream of the retina." and I would suggest rewording it in the abstract.As discussed previously and now nicely added to the discussion, it is difficult to make a direct comparison given the different stimulus types used to characterize the retina and V1 recordings and the different levels of adaptation in both tissues. I will leave this point to the discussion, which allows for a more nuanced description of the phenomenon. Why do I think this is important? In the introduction, the authors argue that "the discrepancy [of previous studies] may be due to differences in stimulus design or light levels." However, while different light levels can be tested in V1, this cannot be done properly in the retina with 2P experiments. To address this, one would have to examine color-opponency in RGC terminals in vivo, which is beyond the scope of this study. Addressing these latter points directly in the discussion would, in my opinion, only strengthen the study.

We thank the reviewer for the feedback. We removed the sentence mentioned by the reviewer from the abstract, as well as from the summary of our results in the Introduction. Additionally, we now phrase the interpretation of the retinal results more conservatively and specifically highlight in the Discussion that comparing ex-vivo retinal to in-vivo cortical data is challenging. With these changes, we believe that the focus of the paper is explicitly defined to be on the neuronal representation of color in mouse visual cortex, rather than on the comparison of retinal and cortical color processing.

Minor:In the abstract, the second sentence says that we already know the mechanisms in primates.Unfortunately, I do not think this is true. First, primates refers to an order with several species, which might have adaptations to their color-processing. Second, I’m aware of several characterizations in "primates" that have led to convincing models (as referenced), but in my opinion, this is far from a true understanding the mechanisms, especially since very little is known about foveal color processing due to the difficulties of these experiments. Similarly in the introduction. "Primates" is indirectly defined as a species. Perhaps some rewording is needed here as well, since we know how different cone distributions can be in rodents (see Peichl’s work).

Thanks. We have reworded the Abstract and Introduction towards indicating that many studies have been performed in primate species, without suggesting that the mechanisms are described.

The legend in Fig. 2 has a "Fig. ???"

Fixed.

**Referee 2 (Remarks to the Author):**
Franke et al. characterize the representation of color in the primary visual cortex of mice, highlighting how this changes across the visual field. Using calcium imaging in awake, head-fixed mice, they characterize the properties of V1 neurons (layer 2/3) using a large center-surround stimulation where green and ultra-violet colors were presented in random combinations. Clustering of responses revealed a set of functional cell-types based on their preference to different combinations of green and UV in their center and surround. These functional types were demonstrated to have different spatial distributions across V1, including one neuronal type (Green-ON/UV-OFF) that was much more prominent in the posterior V1 (i.e. upper visual field). Modelling work suggests that these neurons likely support the detection of predator-like objects in the sky.Strengths: The large-scale single-cell resolution imaging used in this work allows the authors to map the responses of individual neurons across large regions of the visual cortex. Combining this large dataset with clustering analysis enabled the authors to group V1 neurons into distinct functional cell types and demonstrate their relative distribution in the upper and lower visual fields. Modelling work demonstrated the different capacity of each functional type to detect objects in the sky, providing insight into the ethological relevance of color opponent neurons in V1.

We thank the reviewer for appreciating our study.

Weaknesses: While the study presents convincing evidence about the asymmetric distribution of color-opponent neurons in V1, the paper would greatly benefit from a more in-depth discussion of the caveats related to the conclusions drawn about their origin. This is particularly relevant regarding the conclusion drawn about the contribution of color opponent neurons in the retina. The mismatch between retinal color opponency and V1 color opponency could imply that this feature is not solely inherited from the retina, however, there are other plausible explanations that are not discussed here. Direct evidence for this statement remains weak.

Thanks for this comment. We removed the retinal findings from the abstract, as well as from the summary of our results in the Introduction. In addition, we now phrase the interpretation of the retinal results more conservatively and specifically highlight in the Discussion that comparing ex-vivo retinal to in-vivo cortical data is challenging. With these changes, we believe that the focus of the paper is explicitly defined to be on the neuronal representation of color in mouse visual cortex, rather than on the comparison of retinal and cortical color processing.

In addition, the paper would benefit from adding explicit neuron counts or percentages to the quadrants of each of the density plots in Figures 2-5. The variance explained by the principal components does not capture the percentage of color opponent cells. Additionally, there appear to be some remaining errors in the figure legend and labels that have not been addressed (e.g. ’??’ in Fig 2 legend).

Thank you for this suggestion. We believe that adding the numbers or percentages to the figure panels would make them too crowded. Instead, we have now mentioned in the Results section and the legends that the percentages of variance explained by the color (off-diagonal) and luminance axis (diagonal) correlate with the number of neurons located in the color (top left and bottom right) and luminance contrast quadrants (top right and bottom left), respectively. Together with the number of neurons in each plot stated in the legends and the scale bar indicating the number of neurons per gray level, we hope this approach provides clarity for the reader to interpret the panels. Additionally, we have fixed the broken reference in the legend of Fig. 2.

Overall, this study will be a valuable resource for researchers studying color vision, cortical processing, and the processing of ethologically relevant information. It provides a useful basis for future work on the origin of color opponency in V1 and its ethological relevance.General Suggestions:- Please add possible caveats of using ETA method to the discussion section. For example, it is unclear to what extent ON/OFF cells are being overlooked by using ETA method.

We now discuss the limitations of the ETA approach in the Discussion section.

- The caveats of using the percentage of variance explained in the retina as evidence against V1 solely inheriting color-opponency from retinal output neurons are not adequately addressed. For example, could the mismatch in explained variance of the color axis between V1 and RGCs be explained by a subset of non-color opponent RGCs projecting elsewhere (not dLGN-V1) or that color opponent cells project to a larger number of neurons in V1 than non-color opponent cells? We suggest adding a paragraph to the discussion to address this issue.

We have removed these conclusions from the paper, more carefully interpret the retinal results and mention that comparing ex-vivo retina data with in-vivo cortical data is challenging.

- Please clarify how the different response types shown in Figure 5e-f lead to differences in noise detection and thereby differences in predator discriminability. For example, why does Gon/UVoff not respond to the noise scene while Goff/UVoff does?

We added this to the Results section.

- Please clarify the relationship between ETA amplitude, neural response probability, and neural response amplitude. For example, do color-opponent cells have equal absolute neural response amplitudes to the different colors?

Thank you for bringing up this point. The ETA is obtained by summing the stimulus sequences that elicit an event (i.e., response), weighted by the amplitude of the response. Consequently, the absolute amplitude of the ETA correlates with the calcium amplitude. Importantly, the ETA amplitudes of different stimulus conditions are comparable because they were estimated on the same normalized calcium trace. Therefore, comparing the absolute amplitudes of ETAs of color-opponent neurons reveals the response magnitude of the cells to different colors. We have now included this information in the Results section.

Abstract: - "more than a third of neurons in mouse V1 are color-opponent in their receptive field center". It is unclear what data supports this statement. Can you please provide a statement in the manuscript that supports this directly using the number of neurons?

We added the following sentence to the Results section: Nevertheless, a substantial fraction of neurons (33.1%) preferred color-opponent stimuli and scattered along the off-diagonal in the upper left and lower right quadrants, especially for the RF center.

Figure 2: - There is a ?? in the figure legend. Which figure should this refer to? - please provide explicit neuron counts/percentages for each quadrant in b.

We fixed the figure reference. We believe that adding the numbers or percentages to the figure panels would make them too crowded. Instead, we have now mentioned in the Results section and the legends that the percentages of variance explained by the color (off-diagonal) and luminance axis (diagonal) correlate with the number of neurons located in the color (top left and bottom right) and luminance contrast quadrants (top right and bottom left), respectively. Together with the number of neurons in each plot stated in the legends and the scale bar indicating the number of neurons per gray level, we hope this approach provides clarity for the reader to interpret the panels.

Figure 3: - Fig 3: Color scheme makes it very difficult to differentiate the different conditions, especially when printed.

Thanks we changed the color scheme.

- Add explicit neuron counts/percentages for each quadrant in b.

See above.

Figure 4: - Add explicit neuron counts/percentages for each quadrant in b.

See above.

Figure 5: - Add explicit neuron counts/percentages for each quadrant in c.

See above.

Methods: - "we modeled each response type to have a square RF with 10 degrees visual angle in diameter". There appears to be a mismatch between this statement and Figure 5e where 18 degrees is reported.

Thanks we fixed that.

**Referee 3 (Remarks to the Author):**
This paper studies chromatic coding in mouse primary visual cortex. Calcium responses of a large collection of cells are measured in response to a simple spot stimulus. These responses are used to estimate chromatic tuning properties - specifically sensitivity to UV and green stimuli presented in a large central spot or a larger still surrounding region. Cells are divided based on their responses to these stimuli into luminance or chromatic sensitive groups. The results are interesting and many aspects of the experiments and conclusions are well done; several technical concerns, however, limit the support for several main conclusions,Limitations of stimulus choice The paper relies on responses to a large (37.5 degree diameter) modulated spot and surround region. This spot is considerably larger than the receptive fields of both V1 cells and retinal ganglion cells (it is twice the area of the average V1 receptive field). As a result, the spot itself is very likely to strongly activate both center and surround mechanisms, and responses of cells are likely to depend on where the receptive fields are located within the spot(and, e.g., how much of the true neural surround samples the center spot vs the surround region). Most importantly, the surrounds of most of the recorded cells will be strongly activated by the central spot. This brings into question statements in the paper about selective activation of center and surround (e.g. page 2, right column). This in turn raises questions about several subsequent analyses that rely on selective center and surround activation.

Thank you for this comment. A similar point was raised by a reviewer in the first round of revision. We agree with the reviewers that it is critical to discuss both the rationale behind our stimulus design and its limitations to facilitate better interpretation by the reader.

To be able to record from many V1 neurons simultaneously, we used a stimulus size of 37.5 degree visual angle in diameter, which is slightly larger than center RFs of single V1 neurons (between 20 - 30 degrees visual angle depending on the stimulus, see here). The disadvantage of this approach is that the stimulus is only roughly centered on the neurons’ center RFs. To reduce the impact of potential stimulus misalignment on our results, we used the following steps: { For each recording, we positioned the monitor such that the mean RF across all neurons lies within the center of the stimulus field of view.

We confirmed that this procedure results in good stimulus alignment for the large majority of recorded neurons within individual recording fields by using a sparse noise stimulus (Suppl. Fig. 1a-c). Specifically, we found that for 83% of tested neurons, more than two thirds of their center RF, determined by the sparse noise stimulus, overlapped with the center spot of the color noise stimulus.

For analysis, we excluded neurons without a significant center STA, which may be caused by misalignment of the stimulus.

Together, we believe these points strongly suggest that the center spot and the surround annulus of the noise stimulus predominantly drive center (i.e. classical RF) and surround (i.e. extraclassical RF), respectively, of the recorded V1 neurons. This is further supported by the fact that color response types identified using an automated clustering method were robust across mice (Suppl. Fig. 6c), indicating consistent stimulus centering.

Nevertheless, we cannot exclude the possibility that the stimulus was misaligned for a subset of the recorded neurons used in our analysis. We agree with the reviewer that such misalignment might have caused the center stimulus to partially activate the surround. To further address this issue beyond the controls we have already implemented, one could compare the results of our approach with an approach that centers the stimulus on individual neurons. However, we believe that performing these additional experiments is beyond the scope of the current study.

To acknowledge the experimental limitations of our study and the concerns brought up by the reviewer, we have added the steps we perform to reduce the effects of stimulus misalignment in the Results section and discuss the problem of stimulus alignment in the Discussion in a separate section. With this, we believe our manuscript explains both the rationale behind our stimulus design as well as important limitations of the approach.

Comparison with retina A key conclusion of the paper is that the chromatic tuning in V1 is not inherited from retinal ganglion cells. This conclusion comes from comparing chromatic tuning in a previously-collected data set from retina with the present results. But the retina recordings were made using a considerably smaller spot, and hence it is not clear that the comparison made in the paper is accurate. For example, the stimulus used for the V1 experiments almost certainly strongly stimulates both center and surround of retinal ganglion cells. The text focuses on color opponency in the receptive field centers of retinal ganglion cells, but center-surround opponency seems at least as relevant for such large spots. This issue needs to be described more clearly and earlier in the paper.

Thanks for this comment. We removed the retinal findings from the abstract, as well as from the summary of our results in the Introduction. In addition, we now phrase the interpretation of the retinal results more conservatively and specifically highlight in the Discussion that comparing ex-vivo retinal to in-vivo cortical data is challenging. With these changes, we believe that the focus of the paper is explicitly defined to be on the neuronal representation of color in mouse visual cortex, rather than on the comparison of retinal and cortical color processing.

Limitations associated with ETA analysis One of the reviewers in the previous round of reviews raised the concern that the ETA analysis may not accurately capture responses of cells with nonlinear receptive field properties such as On/Off cells. This possibility and whether it is a concern should be discussed.

Thanks for this comment. We now discuss the limitation of using an ETA analysis in the

Discussion section.

Discrimination performance poor Discriminability of color or luminance is used as a measure of population coding. The discrimination performance appears to be quite poor - with 500-1000 neurons needed to reliably distinguish light from dark or green from UV. Intuitively I would expect that a single cell would provide such discrimination. Is this intuition wrong? If not, how do we interpret the discrimination analyses?

Thank you for raising this point. The plots in Fig. 2c (and Figs. 3-5) show discriminability in bits, with the discrimination accuracy in % highlighted by the dotted horizontal lines. For 500 neurons, the discriminability is approx. 0.8 bits, corresponding to 95% accuracy. Even for 50 neurons, the accuracy is significantly above chance level. We now mention in the legends that the dotted lines indicate decoding accuracy in %.